# The Specific Molecular Changes Induced by Diabetic Conditions in Valvular Endothelial Cells and upon Their Interactions with Monocytes Contribute to Endothelial Dysfunction

**DOI:** 10.3390/ijms25053048

**Published:** 2024-03-06

**Authors:** Monica Madalina Tucureanu, Letitia Ciortan, Razvan Daniel Macarie, Andreea Cristina Mihaila, Ionel Droc, Elena Butoi, Ileana Manduteanu

**Affiliations:** 1Biopathology and Therapy of Inflammation, Institute of Cellular Biology and Pathology “Nicolae Simionescu”, 050568 Bucharest, Romania; letitia.ciortan@icbp.ro (L.C.); razvan.macarie@icbp.ro (R.D.M.); andreea.mihaila@icbp.ro (A.C.M.); elena.dragomir@icbp.ro (E.B.); ileana.manduteanu@icbp.ro (I.M.); 2Cardiovascular Surgery Department, Central Military Hospital, 010825 Bucharest, Romania; ionel.droc@gmail.com

**Keywords:** valvular endothelial cells, monocytes, diabetes, endothelial integrity, cytoskeleton, focal adhesion, permeability

## Abstract

Aortic valve disease (AVD) represents a global public health challenge. Research indicates a higher prevalence of diabetes in AVD patients, accelerating disease advancement. Although the specific mechanisms linking diabetes to valve dysfunction remain unclear, alterations of valvular endothelial cells (VECs) homeostasis due to high glucose (HG) or their crosstalk with monocytes play pivotal roles. The aim of this study was to determine the molecular signatures of VECs in HG and upon their interaction with monocytes in normal (NG) or high glucose conditions and to propose novel mechanisms underlying valvular dysfunction in diabetes. VECs and THP-1 monocytes cultured in NG/HG conditions were used. The RNAseq analysis revealed transcriptomic changes in VECs, in processes related to cytoskeleton regulation, focal adhesions, cellular junctions, and cell adhesion. Key molecules were validated by qPCR, Western blot, and immunofluorescence assays. The alterations in cytoskeleton and intercellular junctions impacted VEC function, leading to changes in VECs adherence to extracellular matrix, endothelial permeability, monocyte adhesion, and transmigration. The findings uncover new molecular mechanisms of VEC dysfunction in HG conditions and upon their interaction with monocytes in NG/HG conditions and may help to understand mechanisms of valvular dysfunction in diabetes and to develop novel therapeutic strategies in AVD.

## 1. Introduction

On a worldwide scale, cardiovascular diseases represent a substantial challenge for public health, with aortic valve disease (AVD) emerging as the most prevalent ailment in Western societies [1]. Growing evidence indicates that diabetes constitutes a risk factor for AVD, contributing to disease advancement and leading to an accelerated process [2]. Diabetes is associated with reduced disease prognosis and rapid degeneration of surgically implanted bioprosthetic aortic valves [3]. Research has provided evidence that the prevalence of diabetes is significantly higher among individuals with aortic stenosis, and patients with diabetes experience increased rates of progression from mild to severe aortic stenosis [4]. Open-heart or transcatheter aortic valve replacement is the primary treatment for aortic stenosis, which is one of the most frequent reasons for cardiac surgery. However, both procedures carry a significant risk of adverse events and result in substantial healthcare expenses [5]. To this date, pharmacotherapy has not yet demonstrated the ability to halt disease progression or the subsequent fibrosis process. Clinical trials have shown that conventional cardiovascular drugs, such as statins, have not been effective in slowing down the progression of AVD [6].

The progression of AVD involves a complex cellular-driven process, including endothelial dysfunction, chronic inflammation, endothelial-to-mesenchymal transition (EndMT), monocyte infiltration, oxidative stress, extracellular matrix (ECM) remodeling, switch of valvular interstitial cells (VICs) towards an osteoblastic phenotype, and calcium deposition. These cellular processes contribute to the stiffening, thickening, and altered composition of the valve leaflets, impairing their function [7]. Pioneering research in the field of AVD progression associated with diabetes showed that the valvular endothelium is one of the first vascular territories affected by hyperglycemia (2 weeks after diabetes onset) and that valvular endothelial cells (VECs) transitioned towards a secretory phenotype in a diabetic animal model, exhibiting increased permeability and cytoskeletal modifications [8]. Our recent in vivo studies showed that the aortic valve function and structure were affected early after diabetes onset in ApoE−/− mice (from the first week), showing an increased expression of inflammatory, remodeling, fibrotic, and osteogenic markers. Moreover, we showed that some of the modified molecules were significantly correlated with early valvular dysfunction [9].

Previously considered solely cells with coating functions, VECs are now recognized as a crucial protective barrier against metabolic, mechanical, and inflammatory aggressions and are assumed to have important roles in valve homeostasis and in interactions with cells and molecules from the circulating blood [4]. The regulation of the endothelial barrier function is orchestrated by the quantity and arrangement of intercellular junctions, particularly adherens and tight junctions, actin cytoskeleton organization, and focal adhesions that connect cells to the extracellular matrix. Together, these elements govern the permeability of the valvular tissue [10]. There is evidence that the VEC phenotype is progressively modified in diabetes. The exposure of VEC to high glucose (HG) induces an inflammatory phenotype of VEC, predisposing to monocyte adhesion by mechanisms involving cell adhesion molecules ICAM-1, VCAM-1, E-selectin, and CD18. In addition, HG induced a higher adhesive capacity of VECs compared to aortic endothelial cells, explaining, in part, cardiac valves propensity to accelerated sclerosis in diabetes [11]. It was also shown that chronic exposure to HG induces elevated expression of inflammatory and cell adhesion molecules and changes in the interaction between VECs and the ECM, suggesting alterations in focal adhesion complexes [12,13].

Apart from elevated glucose levels, circulating monocytes could impact the behavior of vascular endothelial cells (VECs). The infiltration of monocytes and the accumulation of monocytes/macrophages in aortic valves are characteristic for AVD [14,15,16]. Recent findings indicate that as much as 15% of cells in a healthy aortic valve have hematopoietic origins, comprising macrophages, T lymphocytes, and B lymphocytes, with their numbers further elevated in calcified valves [17]. While studies on diabetic animal models of atherosclerosis clearly described the contribution of monocytes and macrophages in the atherosclerotic process [18], the specific mechanisms through which diabetes contributes to early valve dysfunction remain unclear. However, it is evident that both VECs and monocytes play crucial roles, and the diabetic environment disrupts cellular homeostasis. Like in other types of endothelia, VEC integrity of the cell monolayer is ensured by the cytoskeleton, junctional complexes, and the adhesions to the extracellular matrix, i.e., the focal adhesions (FAs). In our study, we hypothesize that diabetic conditions or the interplay between monocytes and VECs in diabetic conditions aggravates VEC dysfunction by triggering specific molecular changes in VECs leading to cytoskeleton disorganization, modification in junctional complexes, and in adhesion to ECM, determining increased permeability and enhancement of adhesion and transmigration of monocytes. The objective of this study was to uncover the molecular signatures of VECs in diabetic conditions or upon their interaction with monocytes in NG or HG conditions and to propose novel mechanisms underlying valvular dysfunction in diabetes.

## 2. Results

### 2.1. Transcriptomic Profile of VECs Is Modified by Exposure to High Glucose or by Their Interaction with Monocytes in Normal or High Glucose

In order to identify key genes associated with the pathogenesis of AVD in diabetes in the context of endothelial dysfunction, RNA obtained from VECs cultured in normal glucose (NG), high glucose (HG), or interacted with monocytes pretreated with normal or high-glucose conditions (NGi or Hgi), was analyzed by RNAseq. The sample replicates showed a high degree of correlation, determined by Pearson correlation matrix (Appendix A). After normalizing reads, the Venn diagram showed the differences between gene lists from differential analyses (Appendix A), and the hierarchical clustering analysis showed a distinct profile of interacted VECs compared with noninteracted VECs (Appendix A). The gene ontology (GO) enrichment analysis indicated molecular functions, cellular compartments, and biological processes associated with actin binding, cell-substrate adhesion, membrane rafts, and ECM organization in VECs exposed to high glucose (HG vs. NG). In interacted VECs, GO analysis indicated processes such as actin binding, membrane secretion, cellular activation, and immune cell transport (NGi vs. NG; HGi vs. NG) (Appendix A).

To highlight the role of high glucose exposure or monocyte-VEC crosstalk under normal and diabetic conditions concerning endothelial dysfunction, significantly enriched KEGG related to this process were extracted and visualized using the GOChord package in R [19]. Thus, we visualized differentially expressed genes (DEGs) with a cutoff criteria of an absolute fold change > 1.2 and padj < 0.05 enriched in KEGG pathways: regulation of actin cytoskeleton (hsa04810), PI3K-Akt signaling pathway (hsa04151), focal adhesion (hsa04510), tight junction (hsa04530), adherens junction (hsa04520), cell adhesion molecules (CAMs) (hsa04514), and leukocyte transendothelial migration (hsa04670). VECs under high-glucose conditions (HG) interacted with monocytes under normal glucose (NGi) or interacted with monocytes under high-glucose conditions (HGi) were compared to control VECs, maintained in normal glucose (NG). The total number of up- or downregulated genes in each selected KEGG pathway is presented in bar charts (Figure 1A,C,E). The GOChord visualization reveals DEGs enriched in each pathway and illustrates the connection between pathways considered crucial for maintaining endothelial barrier function. The overall perspective from the GOChord visualization reveals an increased count of genes involved in cytoskeleton regulation, PI3K-Akt signaling, and endothelial transmigration in interacted VECs (irrespective of glucose concentration) compared with genes in uninteracted cells (Figure 1B,D,F).

### 2.2. Exposure of VECs to High Glucose or Co-Culture with Monocytes Induces Specific Modifications to VEC Cytoskeleton

Since our transcriptomic data revealed that upon their crosstalk with monocytes, genes associated with cytoskeleton regulation were found to be modified, we explored the expression of vasodilator-stimulated phosphoprotein (VASP), a conserved actin regulatory protein, and the activation of RhoA/ROCK1 and PI3K-Akt signaling, known to interconnect in the regulation of actin dynamics. Gene expression analysis showed that *VASP* increased in VECs in HG, and in VECs interacted with monocytes in NG and HG conditions (NGi and HGi) compared to normal VECs (NG), and its gene expression decreased significantly (but not to control level) in HGi compared to both HG and NGi. Moreover, *ROCK1* gene expression increased in VECs in HG, NGi, and HGi conditions compared to control VECs (NG), but there was no difference between interacted VECs in NG versus HG conditions. Interestingly, the gene expression of *RhoA*, a molecular switch for the downstream effector ROCK1, was not significantly modified (Figure 2A).

The protein expression of ROCK1 showed a significant increase in VEC in HG, NGi, and HGi conditions compared to control VEC (NG) (Figure 2B), consistent with gene expression analysis. Because PI3K is known to play a role in the assembly and regulation of all major classes of cytoskeletal components, including actin, microtubules, and intermediate filaments [20], and given that our mRNA sequencing data showed the enrichment of PI3K-Akt signaling pathway in VECs employed in our study, we analyzed the activation of p55 and p85 subunits of PI3K. Our data showed that both p55 and p85 are phosphorylated following monocyte interaction, and activation is increased in HG conditions compared with NG conditions (HGi vs. NGi) (Figure 2C,D).

Although regulatory proteins are modulated, α-tubulin protein expression was not modified (Figure 2E). Further, we labeled F-actin and α-tubulin to visualize the cytoskeleton by immunofluorescence. Results showed that uninteracted cells (NG or HG) exhibited a well-defined cytoskeleton characterized by thin stress fibers with a cortical distribution, while interacted cells develop thicker stress fibers in VECs. During cellular interactions, certain VECs showed depolymerized microtubules, and the presence of polymerized α-tubulin along the axis of cell elongation was also observable in adjacent cells, colocalizing with thicker stress fibers (Figure 2F).

### 2.3. High Glucose and the Interaction with Monocytes Induces Integrin Profile Changes in VECs

As disorganization in the cytoskeleton was one of the pathways found to be modified in VECs in HG and upon interaction with monocytes and since our transcriptomic data revealed that VEC in HG or interacted with monocytes in NG or HG conditions exhibit changes in integrins expression profile, we further validated the integrin profile of VECs. Integrins are heterodimeric transmembrane proteins composed of α and β subunits forming receptors for extracellular matrix proteins such as fibronectin, vitronectin, collagen, laminin, and leukocyte-specific receptors. This exploration aimed to gain a deeper understanding of whether modifications occurred in the integration of the cytoskeleton with the ECM within adhesion complexes.

qPCR analysis of integrin subunits showed that VECs in HG exhibited upregulation of integrins α4, α5, and β2 (Figure 3A,B). VEC–monocyte interaction under normal glucose induced the expression of β2 integrin, while the interaction under high glucose decreased the expression of fibronectin, vitronectin, and collagen receptors, namely, α1, α5, and αV, and increased the expression of leukocyte-specific receptors, integrin α4, and integrin β2 (Figure 3A,B).

For the assessment of integrins protein expression, we used the Alpha/Beta Integrin-Mediated Cell Adhesion Array Combo kit (Chemicon). Similar to PCR results, we observed a decrease in the expression of integrins, receptors for fibronectin, vitronectin, and collagen (α1, α5, αV), as well as αVβ5 and α5β1 dimers. Additionally, we observed increased expression of α4 in interacted or noninteracted VECs in high glucose (HG and HGi) and of β2 integrin subunit in interacted VECs in normal or high-glucose conditions (NGi and HGi) (Figure 3C,D).

Furthermore, to assess the functional role of the integrin profile, we conducted an adhesion experiment to ECM proteins (collagen I, IV, fibronectin, vitronectin, laminin) employing VECs in HG or interacted with monocytes in normal or high-glucose concentrations. The results showed that VECs in high glucose and VECs interacted with monocytes in HG were less adhesive to fibronectin and vitronectin, consistent with the observations regarding integrin subunits expression. Also, a decreased adhesiveness to collagen I was observed in VECs upon their interaction with monocytes in HG (Figure 3E).

### 2.4. High Glucose or Interaction with Monocytes in Normal or High Glucose Induces Changes in the Expression of Focal Adhesion Proteins in VECs

Our data showed that the integrin profile was modified by the interaction of VECs with monocytes and further changed in HG conditions compared to normal glucose. Knowing that integrins form mechanical links between the intracellular actin bundles and the ECM, serving as biochemical signaling hubs at specialized sites of cell–ECM adhesions, named focal adhesions (FAs), we analyzed the adaptor and regulator proteins of these multiprotein complexes. Thus, we examined the gene expression of molecules that regulate the formation or disassembly of focal adhesions, such as FAK (focal adhesion kinase) and caveolin-1. Our data showed that under high-glucose conditions, the gene expression of both FAK and caveolin-1 were significantly decreased in VEC interacted with monocytes (HGi) compared to NG, HG, or NGi VECs (Figure 4A). Moreover, Western blot analysis revealed a significant decrease of FAK protein expression in VECs interacting with monocytes (NGi) with significantly lower levels in VECs interacted under high-glucose conditions (HGi) (Figure 4B).

Next, paxillin and vinculin, key components of focal adhesions that play a crucial role in translating extracellular signals into intracellular responses, were evaluated by Western blot. Our data indicated a significant decrease in paxillin in VECs in HG, NGi, or HGi compared to control VECs and a further decrease of paxillin in HGi compared to HG or NGi (Figure 4C). Vinculin expression did not show a significant modulation (Figure 4D). Given that various stimuli have been shown to induce reversible dissipation of vinculin from focal adhesions and internalization of vinculin into vesicles [21], we investigated, by immunofluorescence, the distribution of vinculin in VECs. The same ECM substrates that showed low adherence of VECs, namely, collagen I, fibronectin, or vitronectin, were used for plate coating before cultivating VECs and treating cells with high glucose or direct interaction with monocytes. Our results show that in normal glucose, VECs (NG) and monocyte-interacted VECs (NGi) exhibited vinculin organized in focal adhesions. However, under high-glucose conditions (HG and HGi), vinculin was internalized into the cells, resulting in few observable focal adhesions (Figure 4E).

### 2.5. VECs Exposure to High Glucose or Their Interaction with Monocytes Induces Modifications of Junctional Proteins and Surface Adhesion Molecules

Junctional proteins and cell adhesion molecules play a crucial role in orchestrating intercellular communication and the transfer of signals between cells and their environment. Junctional proteins ensure stable connections between adjacent cells, while cell adhesion molecules facilitate dynamic interactions and transient bonds crucial for processes like immune cell recruitment during inflammation. Our transcriptomic data revealed that VEC and monocytes crosstalk induced modifications involved in cellular communication in VECs. Therefore, we analyzed the gene and protein expression of junctional proteins and surface adhesion molecules in VECs.

The results showed that the molecules *JAM2* (junctional adhesion molecule 2), claudin-5 (*CLDN5*), and cadherins-2, -5, and -11 (*CDH2*, *CDH5*, *CDH11*) exhibited modulations of gene expression under various experimental conditions. Thus, *JAM2* was significantly increased by high-glucose exposure (HG and HGi) and was not directly modulated by monocyte interaction. Claudin-5, a tight junction protein, was increased in VECs exposed to HG, but decreased after VEC interaction with monocytes. The components of adherens junctions, cadherins-2, -5, and -11, were modulated in VECs by monocyte interaction, and only cadherin-2 was significantly decreased in VECs interacted with monocytes in HG conditions (HGi) compared to normal glucose (NGi) (Figure 5A).

Moreover, the gene expression of cell adhesion molecules *ICAM1*, *VCAM1*, *PECAM1*, E-selectin (*SELE*), and *VWF* (von Willebrand factor) was investigated, and only the gene expression of E-selectin was found to be significantly increased in VECs interacted with monocytes in HG conditions (HGi) compared to normal glucose (NG) (Figure 5B).

Further, we validated the protein expression of the observed molecules by Western blot. The protein expression of cadherin-2 was significantly increased in VECs in HG and decreased in monocyte-interacted VECs in normal or high-glucose conditions (Figure 5C). Cadherin-5 was decreased in monocyte-interacted VECs in normal or high-glucose conditions, compared to noninteracted VECs (Figure 5D). The protein expression of E-selectin was significantly increased in VECs interacted in HG conditions (HGi) compared to all other treatments (Figure 5E).

### 2.6. High-Glucose Conditions and the Interaction of VECs with Monocytes in Normal or High Glucose Impacts on Valvular Endothelium Permeability and on Monocyte Adhesion and Transmigration

As modifications in the gene and protein expression of various molecules involved in cell–cell communication and cell–matrix interactions were identified, we investigated the functional role of these changes in cellular permeability. Our findings indicate that the endothelial permeability increased when the endothelium was exposed to monocytes in high-glucose conditions (HGi) compared to control VECs (NG) or VECs in HG. Additionally, when employing specific inhibitors for the two kinases found to be activated, namely, Y27623 to inhibit ROCK and LY294002 to inhibit PI3K, we observed the restoration of permeability to control levels only after inhibiting PI3K, but not ROCK (Figure 6A).

Another functional assay that we performed was the adhesion and transmigration of monocytes. Since there are no studies showing the adhesion process of monocytes pre-exposed to high glucose, VECs maintained in NG or HG culture media were interacted for 2 h with normal-cultured monocytes (Mo-NG) or monocytes pre-exposed to HG (Mo-HG) media. Our findings indicated that the number of monocytes adhered to VECs in HG was higher compared to those that adhered to VEC in NG, and Mo-HG were more adhesive to VECs in HG (Figure 6B).

For transmigration studies, the conditioned media from VECs in NG or HG or VECs interacted with monocytes was employed. The results showed that the number of monocytes migrated towards the conditioned media from VEC-Mo co-culture in HG (HGi) is significantly higher compared to the number of monocytes migrated towards the conditioned media from other conditions, namely, VECs in NG or HG or interacted with monocytes in NG (Figure 6C).

## 3. Discussion

Aortic valve disease (AVD) is a progressive disease with no pharmacological treatment. The prevalence of diabetes among AVD patients is higher than in the general population, with diabetes significantly increasing the risk of AVD development and progression. The interplay between AVD and diabetes-driven mechanisms is not entirely known yet.

In this study, our findings suggest that diabetic conditions, as well as interaction of valvular endothelial cells (VECs) with monocytes in normal or high glucose, have a significant impact on the transcriptomic profile of VECs. The specific molecular changes were identified in processes associated with the regulation of the cytoskeleton, focal adhesions, cellular junctions, and cell adhesion molecules. Exposure of VECs to high glucose led to the modulation of key molecules and regulatory proteins, including increased expression of VASP and ROCK1 (involved in cytoskeleton regulation), enhanced expression of JAM2, cadherin-2, and integrins α4, α5, and β2 (involved in cell–cell and cell–matrix adhesion), and decreased expression of paxillin (an adaptor protein of focal adhesions). Furthermore, the interaction between VECs and monocytes under normal glucose conditions resulted in the upregulation of VASP, ROCK1, integrin β2, and PI3K activation, and downregulation of focal adhesion proteins (FAK and paxillin) and junctional proteins (Claudin-5 and cadherin-5). In high-glucose conditions, this interaction induced a more prominent activation of the PI3K signaling pathway, decreased expression of junctional proteins (FAK, caveolin-1, and paxillin), and upregulation of E-selectin expression. Additionally, the interaction in high-glucose conditions revealed decreased expression of α1, α5, αV, αVβ5, α5β1 integrins, and cadherin-2. These modifications in gene and protein expression led to cytoskeleton disorganization, alterations of junctional complexes, low adherence of VECs to ECM, increased permeability, and enhanced adhesion and transmigration of monocytes. To the best of our knowledge, this study represents the first investigation into the molecular signatures of valvular endothelium in diabetic conditions or during interaction with circulating monocytes.

Valvular endothelial cells were, in the past, thought to serve only as a coating, but they are now recognized as a barrier that protects against metabolic, mechanical, and inflammatory assaults [4]. The complex interaction between various cellular and molecular components contributes to endothelial dysfunction, which can lead to the development of valvular complications. Endothelial dysfunction, a pivotal factor in various cardiovascular and inflammatory disorders, is intricately linked to disruptions in fundamental cellular processes such as cell–cell and cell–matrix communication, cytoskeleton regulation, and normal interaction with monocytes in the bloodstream [22]. Endothelial cells communicate with each other, with circulating blood cells and with the extracellular matrix to maintain vascular integrity and function. Dysregulation in these communication pathways can compromise the barrier function of endothelial cells, leading to increased permeability and susceptibility to inflammatory stimuli [23]. The cytoskeleton, composed of complex protein networks, plays a crucial role in maintaining endothelial cell shape and function; modifications in cytoskeletal regulation can disrupt normal endothelial physiology, impacting vascular permeability [24]. Furthermore, the adhesion of monocytes to the endothelium is a key event in the inflammatory process, and endothelial dysfunction can enhance this adhesion, promoting the infiltration of immune cells into the vascular wall [25]. Understanding the intricate interplay of these processes in the context of aortic valve disease in association with diabetes is essential for unraveling the mechanisms underlying valvular endothelial dysfunction and developing targeted therapeutic strategies.

Monocytes are key components of the immune system, playing a significant role in the progression of AVD and being involved in the inflammatory processes associated with diabetes. They adhere to the valve endothelium, infiltrate into the aortic valve leaflet, and promote inflammation [26]. Monocyte accumulation was shown in diseased valves, and activated monocytes have been found to enhance the expression of proinflammatory and pro-osteogenic factors, inducing fibrotic and osteogenic responses in human aortic valve interstitial cells [27,28]. However, the exact relationship between monocytes and VECs in AVD in the context of diabetes remains to be elucidated.

Integral membrane proteins from the GLUT family, encoded by the SLC2 genes, mediate the transport of glucose in human cells. The endothelium exhibits robust glucose uptake, primarily facilitated by GLUT1 and GLUT3 [29], while paracellular transport through cell–cell junctions is an alternative route for glucose passage [30]. Monocyte glucose uptake also relies mainly on GLUT transporters, with GLUT1 being the most abundant [31]. Our transcriptome sequencing data indicate GLUT1 as the primary glucose transporter in VECs, whereas GLUT4, the insulin-dependent glucose transporter, is not expressed in VECs (sequencing data from repository).

In our study, we focused on exploring VEC dysfunction as a consequence of high-glucose conditions on VECs and of the direct crosstalk with monocytes under normal or diabetic conditions. Our results show that diabetic conditions (HG) and VEC interaction with monocytes in NG and HG impacted the transcriptome profile of VECs. The sequencing data showed differences in gene expression patterns between VECs in high glucose (HG), VECs interacted with monocytes in normal glucose (NGi), and VECs in HG conditions interacted with monocytes pre-exposed to high-glucose concentrations (HGi), and all groups compared with control VECs, maintained in normal glucose (NG). By analyzing the differential expressed genes (DEGs), we identified biological pathways that are enriched, in order to understand the functional implications of the observed gene expression changes. Focusing on pathways that impact endothelial dysfunction, we further analyzed genes related to focal adhesions, cellular junctions, cell adhesion molecules, ECM-receptor binding, and cytoskeleton regulation. We found that all these pathways were modulated in VECs in HG and VECs interacted with monocytes in normal or high glucose, and these transcriptomic signatures might help to gain insights into the complex interplay between cellular and molecular components involved in valvular endothelial dysfunction. Next, we validated certain molecules that are known to play a significant role in the selected pathways.

The cytoskeleton, intercellular junctions, and focal adhesions are closely linked to endothelial dysfunction and play crucial roles in maintaining the endothelial barrier function. The cytoskeleton, composed of actin filaments, microtubules, and intermediate filaments, is vital for the structural homeostasis of endothelial cells and controls the movement of solutes between the bloodstream and tissues. Focal adhesions provide an important structural basis for anchoring the cytoskeleton to the extracellular matrix and regulating cell–matrix adhesion. The intercellular junctions, linked to the actin cytoskeleton, also maintain endothelial integrity. The reorganization of these components significantly impacts endothelial permeability, and understanding their roles in maintaining valvular endothelial integrity has significant implications for the development of potential therapeutic strategies in AVD.

The regulation of the cytoskeleton involves a complex interplay between various signaling molecules and pathways. In our study, we evaluated pathways indicated by the sequencing analysis to modulate cytoskeleton organization, namely, VASP, RhoA/ROCK1, and the PI3K/AKT pathway. The data showed increased expression of ROCK1 (but not RhoA) in all our experimental groups compared to control VECs, and appear to support previous finding regarding ROCK1 signaling involvement in the pathogenesis of various cardiovascular diseases [32]. Additionally, the data indicated the activation of both p55 and p85 subunits of PI3K in VECs interacted with monocytes under diabetic conditions. Moreover, VASP, a downstream effector of ROCK1, was significantly increased in all groups compared to control, its downregulation in HGi compared to HG or NGi conditions, suggesting modifications in cytoskeletal dynamics and cellular adaptation to environmental changes.

The cytoskeleton and integrins are bidirectionally linked in endothelial cells, playing complementary roles in maintaining endothelial integrity. Disruptions in this interplay can lead to pathological conditions such as increased endothelial permeability. Integrins transmit signals from the ECM to the cytoskeleton, influencing cell morphology, adhesion, and migration, and, at the same time, the dynamic rearrangement of the cytoskeleton influences the clustering and activation of integrins, modulating their adhesive properties [33]. In our study, integrin subunits involved in cell adhesion to collagen and proteins containing RGD motif (like fibronectin and vitronectin) were significantly downregulated in VECs interacted with monocytes in HG conditions (HGi). This downregulation led to a decreased ability of HGi-VECs to adhere to collagen I, fibronectin, and vitronectin. Integrins are key components of focal adhesions. The extracellular domain of integrins binds to ECM proteins, while the cytoplasmic tails interact with intracellular proteins associated with focal adhesions, such as paxillin and vinculin, that anchor actin filaments. Numerous investigations have established FAK’s role as a central mediator in integrin signaling and focal adhesion assembly, disassembly, and signaling [34]. In our study, the KEGG pathway “focal adhesion” was significantly enriched in all conditions. Validating FAK’s gene and protein expression, we identified that in VECs interacted with monocytes in HG conditions, FAK was downregulated, possibly explaining the low adhesiveness of these cells to different ECM substrates. More evidence supports the hypothesis that caveolin-1, a protein involved in cell motility and cytoskeletal organization, also regulates focal adhesion turnover and FAK stabilization [35,36]. Thus, in our experimental conditions, CAV1 gene expression was significantly decreased in HGi-VECs. Moreover, the recruitment of activated PI3K to focal adhesions may lead to subsequent activation and downstream signaling influencing cell adhesion. Analyzing intercellular adapter proteins involved in focal adhesion formation, we observed a significant decrease in the expression of paxillin in all groups compared to the control cells, but a more accentuated downregulation in interacted VECs in high glucose; also, we showed the internalization of vinculin into vesicles under high-glucose conditions. These findings demonstrate the dynamic regulation of focal adhesion formation and disassembly in VECs, influenced by both the glucose concentration in the media and the direct interaction with diabetic monocytes.

Endothelial barrier function and vascular permeability are governed by the number and organization of intercellular junctions, namely, tight and adherens junctions [10]. Endothelial tight junctions, composed of claudins, junctional adhesion molecules (JAMs), occludin, and other adhesion molecules, are regulated by signaling pathways including PKC, RhoA, MAPK, and PI3K-Akt [37]. JAM proteins have been linked with cardiovascular diseases due to their physical and functional interactions with integrins, mediating transient interactions between leukocytes and endothelial cells [38]. Moreover, JAM-A has been proposed as a marker of acute endothelial activation and dysfunction [39]. Claudin-5 is another tight junction protein with a crucial role in maintaining the integrity of the endothelial barrier and contributes to the regulation of paracellular permeability [40,41]. In our study, JAM2 gene expression was significantly increased in VECs by high-glucose conditions, but was not influenced by the interaction of VECs with monocytes. Moreover, claudin-5 was found to be downregulated in VECs interacted with monocytes, disregarding the glucose concentration, indicating that VECs interaction with monocytes affects tight junction integrity in VECs.

In addition to tight junctions, adherens junctions control the paracellular permeability to circulating leukocytes and solutes, as they are located at cell-to-cell contacts and mediate cell adhesion while transferring intracellular signals [42,43]. The actin cytoskeleton plays a crucial role in the organization and stability of adherens junctions, as actin stress fibers in adjacent endothelial cells are linked through adherens junctions [44]; thus, the cytoskeleton regulation and adherens junction integrity are closely linked. In our study, cadherin-2, cadherin-5, and cadherin-11 are significantly decreased in VECs following their interaction with monocytes, suggesting that VECs interaction with monocytes affects tight junction integrity in VECs.

Endothelial permeability denotes the capacity of endothelial cells to allow the passage of molecules between the bloodstream and tissues. Certain pathological conditions, including hyperglycemia, may result in increased endothelial permeability, enabling an excessive flow of fluids and proteins into adjacent tissues, leading to inflammation. The imbalanced regulation of endothelial permeability plays a role in numerous conditions and has the potential to impact both the morbidity of diseases and their treatment outcomes [10]. Therefore, precise regulation of endothelial permeability is crucial for maintaining homeostasis and the normal function of the vascular system.

Although in diabetes, vascular endothelial permeability is well known to be increased due to various factors, including hyperglycemia and free radicals production, leading to the development of vascular complications [45,46], there is limited information available on endothelial permeability in aortic valves due to diabetes. Few studies have investigated the homeostatic barrier function in valvular endothelial cells (VEC), and it was shown that the valve endothelium had higher permeability to LDL than the aorta and arteries [47]. Another study modeled the valve leaflet as a noncontinuous monolayer with leaky cells dispersed and a basement intimal lining, and found evidence of dispersed leaks in vitro. These unique transport and lipid diffusion properties demonstrate a distinct feature of valvular barrier function and subendothelial space [48]. Our data indicate a significant increase in the permeability of valvular endothelium, particularly following the interaction between VECs and monocytes in diabetic conditions. This increased permeability is restored to control levels upon inhibiting PI3K, suggesting the involvement of this signaling pathway in the regulation of endothelial barrier function. The findings align with previous data demonstrating the role of PI3K in vascular permeability [49]. However, further research is needed to fully understand the endothelial permeability in aortic valves.

The adhesion of circulating monocytes to endothelial cells is considered one of the earliest events in atherogenesis and is mediated by the interaction of adhesion molecules on endothelial cells with their integrin counterreceptors on monocytes. The initial rolling of monocytes along the activated endothelium results in a firm adhesion that influences their transmigration [25]. Endothelial junction integrity is essential for the regulation of monocyte adhesion and transmigration, and it is regulated by junctional adhesion molecules that constitute closely associated tight and adherens junctions [50]. It has been shown that elevated glucose concentrations induce monocyte adhesion to valvular endothelial cells [11]. In addition, our results suggest that monocytes pre-exposed to high glucose exhibit increased adhesion to VECs in HG. Furthermore, a significant increase in the number of monocytes migrating towards the conditioned media from VEC-Mo co-culture in high glucose is observed. These findings imply a possible link between elevated glucose levels and altered monocyte-VEC interactions, which could have implications for understanding inflammatory processes associated with diabetic environment in the aortic valve.

This study has some limitations that need to be addressed in future studies. Although the complex diabetic milieu includes high glucose levels, fatty acids, lipid accumulation, insulin resistance, increased oxidative stress and inflammatory cytokines expression, the present study investigated the influence of hyperglycemia as the primary factor responsible for the onset of diabetic complications affecting the aortic valve. Moreover, the co-culture model employed may have certain limitations. Further studies involving co-culture models incorporating valvular endothelial cells, interstitial cells, and monocytes could offer greater relevance in elucidating the mechanisms underlying aortic valve disease associated with diabetes.

In conclusion, our study revealed novel mechanisms of valvular endothelium dysfunction in diabetic conditions and showed that the barrier function of valvular endothelial cells is compromised by diabetic conditions and by VECs interaction with monocytes. To the best of our knowledge, we identified, for the first time, specific molecular changes in VECs at the level of cytoskeleton, junctional proteins, and focal adhesion proteins. These specific changes led to cytoskeleton disorganization, modification in junctional complexes and determined low adherence of VECs to ECM, increased permeability, and enhancement of adhesion and transmigration of monocytes. Thus, we provide novel data on the molecular signatures of VECs in diabetic conditions or upon their interaction with monocytes in NG or HG conditions, and we propose novel molecular mechanisms which could operate in valvular dysfunction in diabetes. Moreover, our results might help to develop novel therapeutic strategies for valve dysfunction in diabetes.

## 4. Materials and Methods

### 4.1. Cell Culture

Primary human valvular endothelial cells (VECs) were sourced from aortic valve leaflets extracted from patients undergoing aortic valve replacement surgery, with either aortic valve regurgitation (characterized by inadequate closure of the valve leaflets, without calcifications) or calcific aortic valve disease. Leaflet areas lacking visible calcifications were selected and further processed for VEC isolation. Samples were collected with the consent of patients, according to the protocol of Dr. Carol Davila Central Military Emergency University Hospital and the principles regarding human sample use in experiments outlined in the Declaration of Helsinki. After removing calcified regions from valve cusps, the normal leaflet tissue underwent a brief wash in phosphate-buffered saline (PBS) and subsequent enzymatic digestion using collagenase I (Merck, Darmstadt, Germany). Cells obtained through this process were centrifuged and seeded into 24-well culture plates coated with 1% gelatin from porcine skin (Sigma-Aldrich, Taufkirchen, Germany), then cultured in EGM-2 Endothelial Cell Growth Medium (Lonza, Basel, Switzerland) supplemented with 10% fetal bovine serum (FBS) and 1% penicillin/streptomycin (Gibco, Grand Island, NY, USA) until reaching confluence. Given the mixed nature of the cell population, VECs were isolated by employing CD31+ magnetic beads (Miltenyi Biotec, Bergisch Gladbach, Germany).

Monocyte-like cell line THP-1 was grown in suspension in the RPMI 1640 culture medium containing 10% FBS and was split 1:5, twice a week.

### 4.2. Cell Treatments

The isolated VEC population was cultured in EGM-2 media supplemented with 10% FBS and 1% penicillin/streptomycin in normal glucose concentration (NG—5 mM glucose) or in high-glucose concentrations (HG—33 mM glucose). The high-glucose concentration has been extensively employed in previous research involving aortic endothelial cells, as well as vascular or cerebral endothelial cells of human, porcine, or murine origin [51,52,53]. To establish the VEC-monocyte co-culture, 1.5 × 10^6^ monocytes/mL were added to the VEC monolayer for a duration of 2 h. The interaction was established between VECs and monocytes in NG media or between VECs cultured in HG and monocytes pre-exposed to HG for 72 h.

### 4.3. RNA Isolation and Transcriptome Sequencing

To profile the gene expression of VECs after interaction with monocytes in NG or HG conditions, monocytes were removed by washing with PBS, and total RNA from VECs was isolated with TRIzol reagent and Phasemaker tubes (ThermoFisher, Waltham, MA, USA). VECs in NG or HG media were used as controls. For each condition, 3 biological replicates were collected in 3 independent experiments. Total RNA was quantified using NanoDrop 1000 (ThermoFisher). Library construction, quality control, and sequencing were performed by Novogene (Cambridge, UK). A total amount of 1 µg RNA per sample was used as input for RNA analysis. Messenger RNA was purified from total RNA using poly-T oligo-attached magnetic beads. After fragmentation, the first-strand cDNA was synthesized using random hexamer primers, followed by the second-strand cDNA synthesis using either dUTP for directional library or dTTP for nondirectional library. Sequencing libraries were generated using the NEBNext UltraTM RNA Library Prep Kit for Illumina following the manufacturer’s recommendations, and index codes were added to attribute sequences to each sample. The library was checked with Qubit and real-time PCR for quantification and bioanalyzer for size distribution detection. The clustering of index-coded samples was performed, and after cluster generation, the quantified libraries were pooled and sequenced using Illumina NovaSeq 6000 and paired-end reads were generated. Raw data of fastq format were processed through fastp software (version 0.23.1) [54]. Clean data were obtained by removing reads containing adapter and poly-N sequences and low-quality reads. Q20, Q30, and GC content of the clean data are presented in Appendix A.

All the downstream analyses were based on the clean data with high quality. Index of the reference genome was built using Hisat2 v2.0.5 and paired-end clean reads were aligned to the reference genome using Hisat2 v2.0.5. For fusion analysis, Starfusion software (1.9.0) was used. The mapping result summary is presented in Appendix A.

For quantification of gene expression levels, featureCounts v1.5.0-p3 was used to count the reads numbers mapped to each gene. FPKM (expected number of fragments per kilobase of transcript sequence per millions base pairs sequenced) of each gene was calculated based on the length of the gene and reads count mapped to this gene, and the data were used for the PCA and Pearson correlation coefficient matrix (Appendix A).

Differential expression analysis was performed using the DESeq2 R package (1.20.0). The resulting *p*-values were adjusted using the Benjamini and Hochberg’s approach for controlling the false discovery rate. Genes with an adjusted *p*-value ≤ 0.05 were assigned as differentially expressed. For the enrichment analysis of differentially expressed genes, the clusterProfiler R package (version 4.10.0) was used to test the statistical enrichment of differential expression genes in KEGG pathways.

### 4.4. Real-Time PCR Analysis

Validation of key molecules found to be modified by RNA-seq was performed by qPCR using RNA obtained from pooled VECs isolated from subsequent experiments. Total cellular RNA was extracted from VEC_NG, VEC_HG, VEC_NGi, and VEC_HGi using TRIzol (Thermo Fisher, Waltham, MA, USA) or Qiagen PureLink RNA Kit (Ambion, Austin, TX, USA). First-strand cDNA synthesis was performed employing 1 μg of total RNA and MMLV reverse transcriptase, according to the manufacturer’s protocol (Invitrogen, Waltham, MA USA). Assessment of mRNA expression was carried out by amplification of cDNA using a LightCycler 480 Real-Time PCR System (Roche, Basel, Switzerland) and SYBR Green I. The primer sequences for the mRNAs of interest are shown in Appendix A. The relative quantification was carried out using the comparative CT method and expressed as arbitrary units. β2 microglobulin (B2M) was used as a reporter gene.

### 4.5. Protein Isolation and Western Blot Analysis

Following interaction, VECs were washed using cold HBSS. Cells were lysed using RIPA lysis buffer supplemented with a protease inhibitor cocktail. After centrifugation (12,000× *g*), the proteins were quantified by bicinchoninic acid (BCA) Protein Assay Kit. Samples (30 μg protein) were separated on 8–12% SDS-PAGE (sodium dodecyl sulfate-polyacrylamide) gel electrophoresis and transferred to nitrocellulose membranes, which were subsequently probed with specific antibodies for ROCK1 (Thermo Scientific, Waltham, MA, USA), phospho-p85/phospho-p55 (Cell Signaling Technology, Danvers, MA, USA), PI3K, (Abcam, Cambridge, UK), α-tubulin (Thermo Scientific), p-FAK (Cell Signaling Technology), FAK (Cell Signaling Technology), paxillin (Cell Signaling Technology), vinculin (Thermo Scientific), cadherin-2 (Thermo Scientific), cadherin-5 (Thermo Scientific), E-selectin (Thermo Scientific), β-actin (Sigma-Aldrich, Saint Louis, MO, USA), and β-tubulin (Abcam). The signals were visualized using SuperSignal West Pico chemiluminescent substrate (Pierce, Appleton, WI, USA) and quantified by densitometry employing the gel analyzer system Luminescent image analyzer LAS 4000 (Fujifilm, Freiburg, Germany) and the Image reader ImageQuant™ LAS 4000 software (GE Healthcare, Chicago, IL, USA).

### 4.6. Identification of Integrins Profile on the Surface of Valvular Endothelial Cells

To determine the integrin profile on the VECs surface, the Chemicon Alpha/Beta Integrin-Mediated Cell Adhesion Array Combo Kit (Merck Millipore, Burlington, MA, USA) was used. This kit represents an efficient method for identifying integrins expressed on the cell membrane. The kit utilizes monoclonal antibodies for human integrins: α1, α2, α3, α4, α5, αV, αVβ3, β1, β2, β3, β4, β6, αVβ5, α5β1, immobilized on two 96-well culture plates. Prior to cell seeding, the strips are rehydrated with PBS for 10 min at room temperature. VECs were dissociated by nonenzymatic digestion (PBS + 5 mM EDTA), and 1 × 10^6^ cells/mL from the cell suspension were resuspended in HBSS buffer + 0.1% BSA, 25 mM HEPES, and 1 mM Ca2+/Mg2 (Assay Buffer). The cell suspension from three biological replicates was incubated on the integrin plates for 2 h at 37 °C. After incubation, cells were washed three times with Assay Buffer and incubated with 4× Cell Lysis Buffer and CyQuant GR for 15 min. The resulting solution was transferred to a black plate, and fluorescence was measured at 485/530 nm.

### 4.7. Immunofluorescence

Cultured VECs in different experimental conditions were washed with cold HBSS, fixed with 4% paraformaldehyde, and blocked with 3% bovine serum albumin (BSA). Cells were incubated overnight with primary anti-human antibodies for α-tubulin and vinculin, followed by incubation with AlexaFluor594-conjugated secondary antibodies. Nuclei were stained with 4′,6-diamidino-2-phenylindole (DAPI), and F-actin was labeled with fluorescein-conjugated or AlexaFluor594-conjugated phalloidin. Images were acquired using the fluorescence microscope Olympus IX8 (Olympus, Tokyo, Japan) equipped with an XM10 camera and processed using ImageJ software (version 1.54f).

### 4.8. Cell Adhesion to Extracellular Matrix Proteins

To determine VECs adhesion to extracellular matrix (ECM) proteins, culture plates were coated with proteins such as collagen I (10 µg/cm^2^), collagen IV (10 µg/cm^2^), fibronectin (5 µg/cm^2^), vitronectin (50 ng/cm^2^), laminin (2 µg/mL), 0.01% gelatin, and BSA (10 µg/mL). After a 3 h incubation, the protein solution was removed, and the plates were air-dried overnight. Before cell incubation on the coated plates, each well was blocked for 1 h with PBS + 0.5% BSA. VECs were dissociated with 0.5% trypsin and transferred to the culture plate coated with different ECM proteins. After 20 min, cells were washed with PBS + 0.1% BSA and fixed with 4% paraformaldehyde. Subsequently, cells were stained with 0.1% Crystal Violet solution + 10% ethanol, washed with distilled water, solubilized with 2% SDS, and the colorimetric reading was measured at 550 nm.

### 4.9. Permeability Assay

VECs were cultured in Transwell chambers on membrane filters with 0.4 µm pores. Upon reaching confluence, they were interacted, for 2 h, with monocytes in normal glucose (NG) or high glucose (HG) culture medium. The medium in the lower compartment of the Transwell chamber was removed and replaced with phenol-red-free RPMI medium. Cells in the upper compartment were incubated with phenol-red-free RPMI medium containing HRP (horseradish peroxidase). After 5 min, the membrane filters were removed, and TMB substrate was added to the medium in the lower compartment. The reaction was stopped with 2N H_2_SO_4_, and the absorbance was measured at 450 nm, with correction at 540 nm.

### 4.10. Monocyte Adhesion and Transmigration

THP-1 monocytes were cultured for 72 h in media with normal (5 mM) or high (33 mM) glucose and labeled with 10 μmol/L of the fluorescent dye 2′7′-bis(2-carboxyethyl)-5(6)-carboxyfluorescein acetoxymethyl ester at 37 °C for 1 h in RPMI 1640 culture medium and subsequently washed by centrifugation. Confluent VECs cultured in NG or HG media were incubated with 10^6^ cells/mL at 37 °C for 1 h. Nonadherent cells were removed by washing with warm culture medium and fixed with 4% paraformaldehyde. Fluorescent monocytes adhered to VECs were counted in three random fields under fluorescence microscopy (Olympus IX81). For transmigration assay, THP-1 cells in normal glucose were added in the upper compartment of a transwell system (8 μm pore size), containing in the lower compartment conditioned media from VEC in normal or high glucose, or previously interacted with monocytes for 2 h in normal or high-glucose conditions. The transwell system was incubated for 2 h at 37 °C, then migrated monocytes present in the lower compartment were counted in three random fields using a phase-contrast inverted microscope Olympus IX81 (Olympus, Tokyo, Japan) and cellSens software (version 4.2). 

### 4.11. Statistical Analysis

The data are expressed as the mean of at least three different experiments ± standard error (SE). Statistical analysis was performed using GraphPad Prism version 9.0.0 software. The *p*-value for comparisons between different samples was determined using Student’s *t*-test, one-way ANOVA, and Bonferroni post hoc test (samples were considered significantly different for *p*-values < 0.05).

### 4.12. Data Access

All sequencing data have been deposited in the ArrayExpress database, https://www.ebi.ac.uk/biostudies/arrayexpress/studies/E-MTAB-13766?key=a37d059b-e285-4b4b-b697-fa4a47c92bcc (accessed on 6 February 2024). All other data are available from the authors on request.

## Figures and Tables

**Figure 1 ijms-25-03048-f001:**
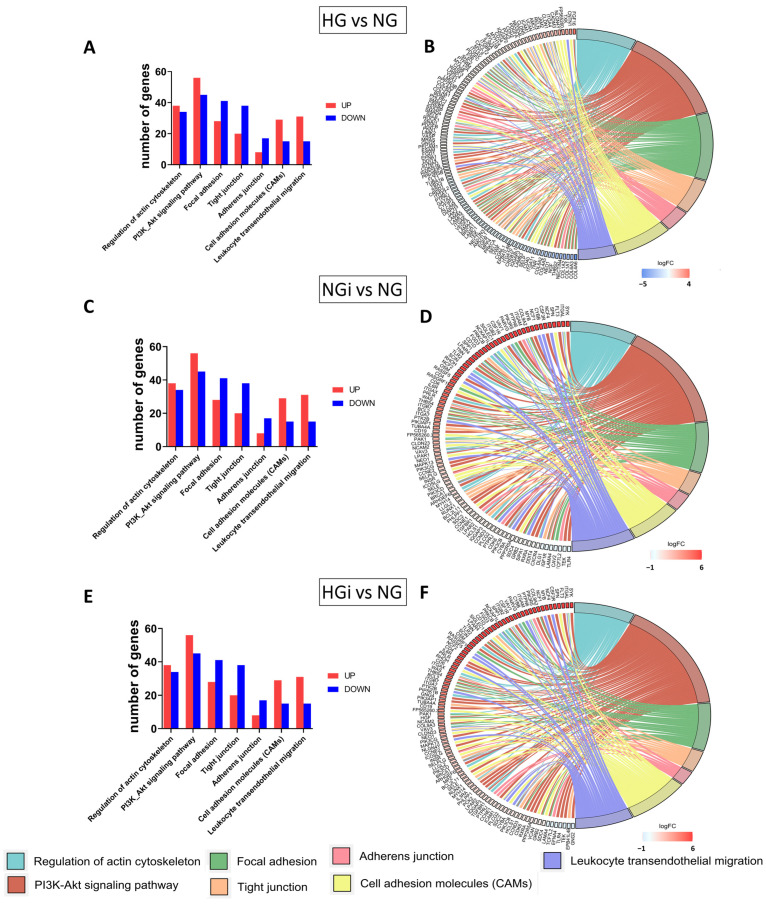
Differentially expressed genes (DEGs) associated with enriched KEGG pathways. DEGs with fold change > 1.2 and padj < 0.05 in VECs under high-glucose conditions (HG) vs. control VECs (NG) (**A**,**B**), interacted with monocytes under normal glucose (NGi) vs. NG (**C**,**D**) or interacted with monocytes under high-glucose conditions (HGi) vs. NG (**E**,**F**) were analyzed. Bar charts and chord plots illustrate the connections between genes and KEGG pathways: Regulation of actin cytoskeleton, PI3K-Akt signaling pathway, focal adhesion, tight junction, adherens junction, cell adhesion molecules (CAMs), and leukocyte transendothelial migration.

**Figure 2 ijms-25-03048-f002:**
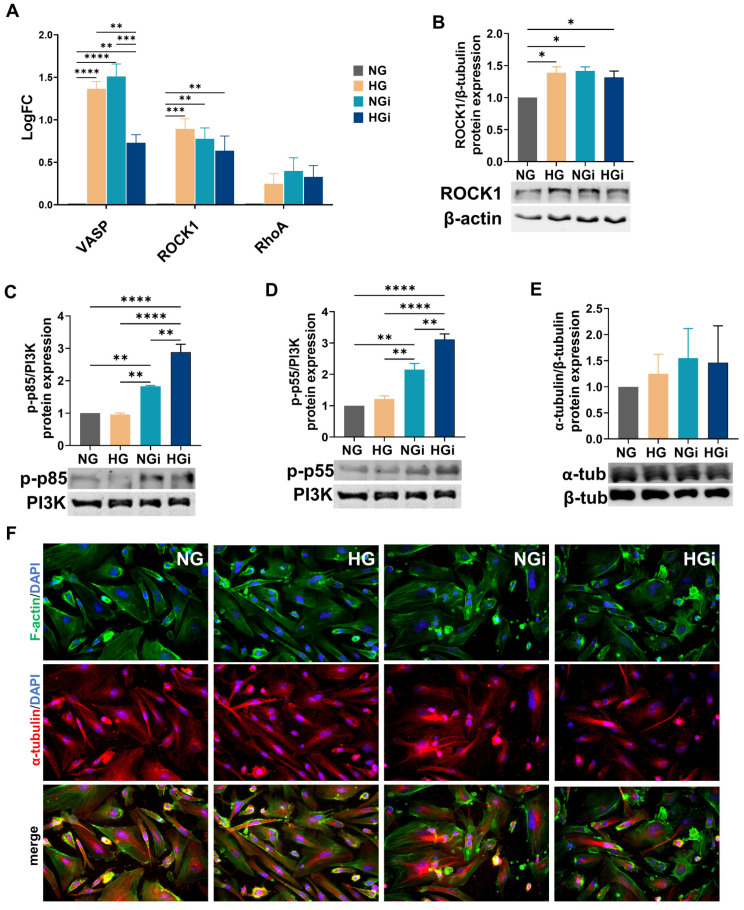
High glucose or interaction of VECs with monocytes modulates VECs cytoskeleton. (**A**) Gene expression analysis of VASP, ROCK1, and RhoA in VECs in normal (NG) or high glucose (HG) in interaction with monocytes (i). (**B**) Protein expression of ROCK1 normalized to β-actin protein level; (**C**,**D**) the phosphorylation of PI3K subunits p55 and p85 normalized to PI3K protein expression; (**E**) protein expression of α-tubulin normalized to β-tubulin. (**F**) The distributions of F-actin fibers (upper panel) and α-tubulin (middle panel) were assessed by immunofluorescence using FITC-phalloidin (for F-actin, green) or a specific primary antibody for α-tubulin and Alexa594-coupled secondary antibody (red) on cultured VEC in normal glucose (NG), high-glucose conditions (HG), or interacted with monocytes in NG (NGi) or HG media (HGi). Nuclei were stained with DAPI (blue). Magnification objective 20×. For graphs in (**A**–**D**), *n* ≥ 3, * *p* < 0.05, ** *p* < 0.01, *** *p* < 0.001, **** *p* < 0.0001 using two-way ANOVA (**A**) or one-way ANOVA (**B**–**E**).

**Figure 3 ijms-25-03048-f003:**
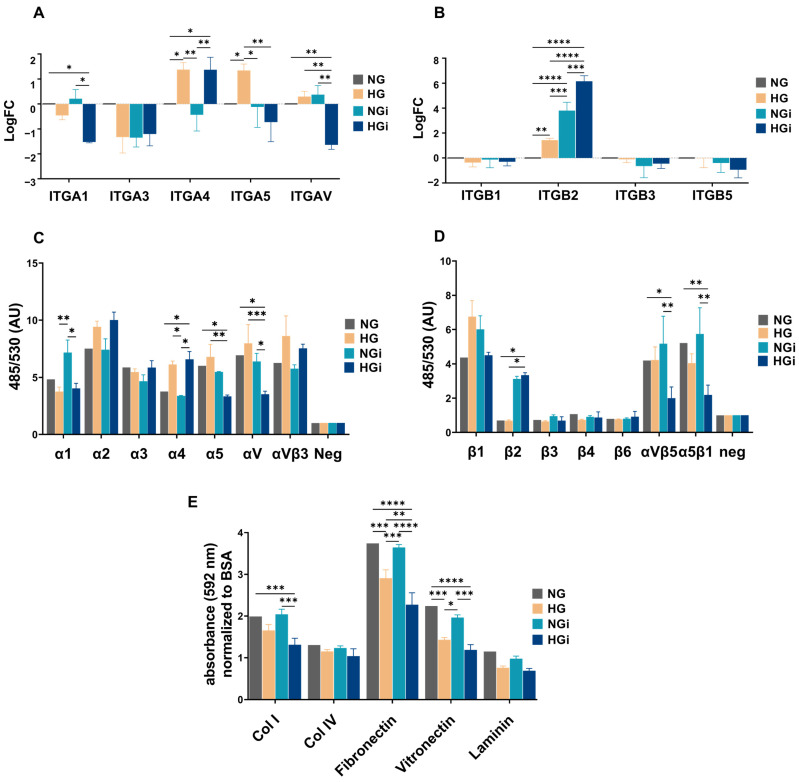
Exposure of VEC to high glucose or their interaction with monocytes induces alterations in the expression of integrins in VECs, leading to a diminished adhesiveness of the endothelium to ECM components. (**A**,**B**) Gene expression analysis of α subunits (**A**) and β subunits (**B**) of integrins in VECs in normal (NG) or high glucose (HG) in interaction with monocytes (i). The mRNA of integrin subunits was normalized to β2-microglobulin mRNA and the data are represented as logFC (log2 fold-change) over control VECs (NG). (**C**,**D**) The integrin expression level was assessed in VECs NG, HG, NGi, and HGi by using the Alpha/Beta Integrin-Mediated Cell Adhesion Array Combo Kit. Values were normalized to negative control from three independent experiments. (**E**) VEC adhesivity to ECM proteins: collagen I, collagen IV, fibronectin, vitronectin, and laminin. *n* ≥ 3, * *p* < 0.05, ** *p* < 0.01, *** *p* < 0.001, **** *p* < 0.0001 using 2-way ANOVA.

**Figure 4 ijms-25-03048-f004:**
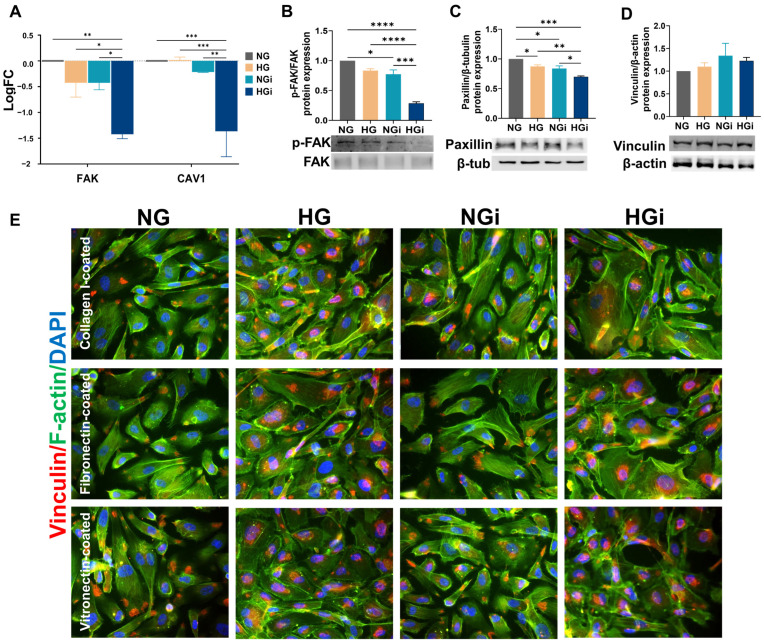
High glucose induces focal adhesion remodeling in valvular endothelial cells. (**A**) Gene expression analysis of focal adhesion molecule (FAK) and caveolin-1 (CAV1) in VECs in normal (NG) or high glucose (HG) in interaction with monocytes (i). The mRNA of focal adhesion molecules was normalized to β2-microglobulin mRNA and the data are represented as logFC (log2 fold-change) over control VECs (NG). (**B**) Protein expression of phosphorylated FAK normalized to unphosphorylated FAK; (**C**) Paxillin protein expression normalized to β-tubulin; (**D**) Vinculin protein expression normalized to β-actin. (**E**) The expression of vinculin was also evaluated by immunofluorescence using a specific primary antibody and Alexa594-coupled secondary antibody (red) on cultured VEC in normal glucose (NG), high-glucose conditions (HG), or interacted with monocytes in NG (NGi) or HG media (HGi). Nuclei were stained with DAPI (blue) and F-actin was stained with FITC-phalloidin (green). Magnification objective 20x. For graphs in (**A**–**D**), *n* ≥ 3, * *p* < 0.05, ** *p* < 0.01, *** *p* < 0.001, **** *p* < 0.0001 using two-way ANOVA (**A**) or one-way ANOVA (**B**–**D**).

**Figure 5 ijms-25-03048-f005:**
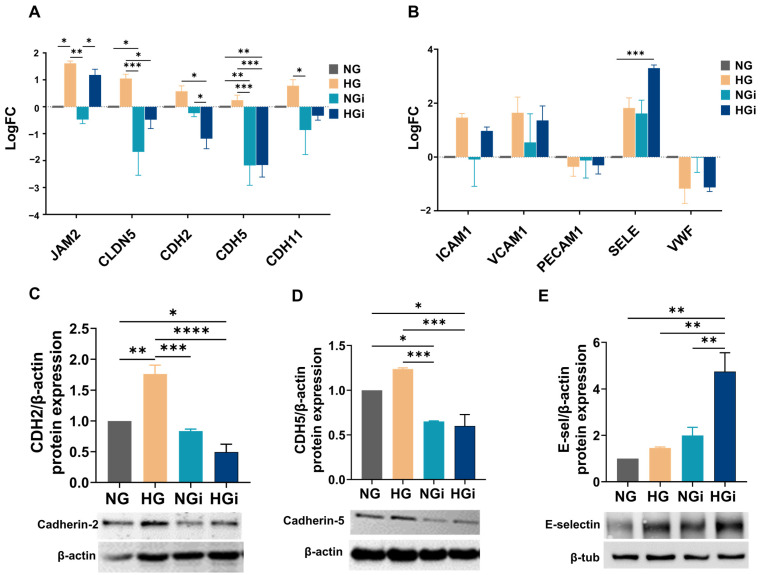
High-glucose conditions and the interaction of VECs with monocytes in normal and high glucose modulates the expression of molecules involved in cellular communication. (**A**) Gene expression analysis of *JAM2*, claudin-5 (*CLDN5*), cadherin-2, -5, and -11 (*CDH2*, *CDH5*, *CDH11*) in VECs in normal (NG) or high glucose (HG) in interaction with monocytes (i). (**B**) Gene expression analysis of *ICAM-1*, *VCAM-1*, *PECAM-1*, E-selectin (*SELE*), and von Willebrand factor (*VWF*). The mRNA of junctional proteins and cell adhesion molecules was normalized to β2-microglobulin mRNA and the data are represented as logFC (log2 fold-change) over control VECs (NG). (**C**–**E**) Protein expression of cadherin-2, cadherin-5 and E-selectin were normalized to β-actin and β-tubulin protein level. *n* ≥ 3, * *p* < 0.05, ** *p* < 0.01, *** *p* < 0.001, **** *p* < 0.0001 using two-way ANOVA (**A**,**B**) or one-way ANOVA (**C**–**E**).

**Figure 6 ijms-25-03048-f006:**
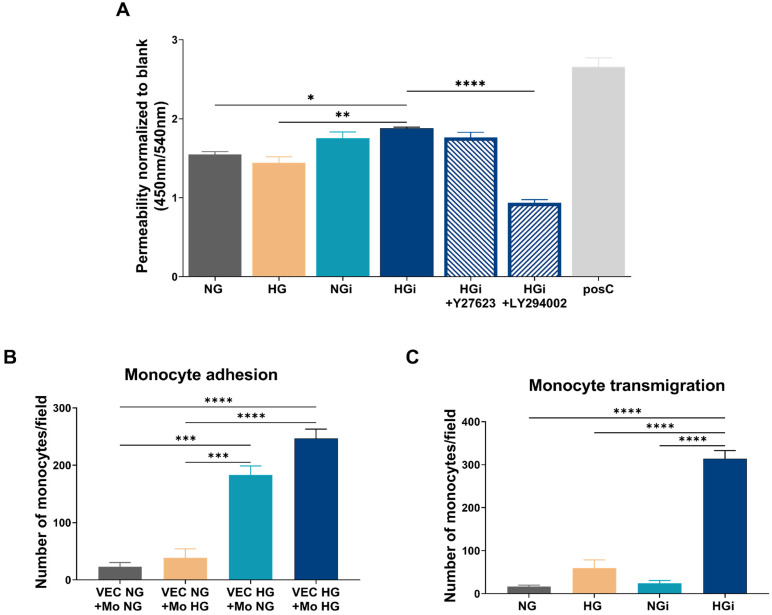
Valvular endothelium function is altered by high glucose or by interaction with monocytes. (**A**) HRP permeability of the VEC monolayer barrier in normal glucose (NG), high glucose (HG), or after interaction with monocytes (i) under NG or HG conditions; Y27623—inhibitor for ROCK; LY294002—inhibitor for PI3K; positive control (posC) represent the passage of HRP through transwell membrane without endothelial layer. (**B**) The adhesion of normal-glucose- or high-glucose-cultured monocytes to VECs maintained in normal or diabetic conditions. (**C**) The transmigration of monocytes towards conditioned media from VECs NG, HG, NGi or HGi. *n* ≥ 3, * *p* < 0.05, ** *p* < 0.01, *** *p* < 0.001, **** *p* < 0.0001 using one-way ANOVA.

## Data Availability

The datasets presented in this study can be found in online repositories. The names of the repository/repositories and accession number(s) can be found here: https://www.ebi.ac.uk/biostudies/arrayexpress/studies/E-MTAB-13766?key=a37d059b-e285-4b4b-b697-fa4a47c92bcc. All other data are available from the authors on request.

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
