# Peer review of "The Specific Molecular Changes Induced by Diabetic Conditions in Valvular Endothelial Cells and upon Their Interactions with Monocytes Contribute to Endothelial Dysfunction"

_ijms, 2024, doi:10.3390/ijms25053048_

Round 1

Reviewer 1 Report

Comments and Suggestions for Authors

The present experimental study is intriguing. However, the study is the first step in exploring Valvular Endothelial Cells and Their Interactions with Monocytes contributing to Endothelial Dysfunction. It is well known from the literature that the immune system plays a central role in many processes of age‐related disorders and Diabetes Mellitus. Functional characterization revealed an insulin‐driven immunometabolism network that supports multiple aspects of phagocytosis. Such reprogramming is associated with a skewed trend of DNA demethylation at the promoter regions of various phagocytic genes as a direct transcriptional effect induced by nuclear‐localized insulin receptors. Together, these highlighted that preserving insulin sensitivity is a key to a healthy lifespan. In monocytes, phagocytic genes are targets of insulin receptors and insulin‐regulated transcription factors.

In the present study, the researchers investigated the role of hyperglycemia only. However, the diabetic milieu includes lipids and a small amount of insulin in DM type 2 individuals with hyperglycemia.

The authors should add a paragraph with the limitations mentioned above.

Furthermore, the authors should describe the mechanism of glucose passage into Endothelial Cells and monocytes. Is insulin necessary, or can glucose enter Endothelial Cells and monocytes freely?

The methodology is appropriate. The manuscript is well written, and the discussion/conclusions are acceptable.

Overall, the data are of interest.

Comments on the Quality of English Language

none

Reviewer 2 Report

Comments and Suggestions for Authors

Tucureanu et al. attempted to investigate potential mechanisms of valvular endothelium dysfunction using valvular endothelial cells (VECs) alone or in co-culture with monocytes (VECis) under normal and high glucose. The authors demonstrated significant changes in the ECM cytoskeleton, junctional proteins, cell permeability, and focal adhesion proteins. 

Comments:

1/ Abbreviation for NG (normal glucose) should be defined in the abstract.

2/ Authors mentioned that primary VECs were sourced from aortic valve leaflets extracted from patients undergoing aortic valve replacement surgery. However, samples may be altered by the presence of macrocalcification-associated inflammation and possible structural changes in ECM. How do authors overcome challenges to secure good quality sample preparation for consistency and reproducibility of experiments?

3/ Did the authors attempt to study the effect of glucose dose-response and treatment duration?

4/ Data reported for Fig.1A, Fig.1C & Fig.1E seems to be similar regardless of the experimental conditions. Any explanation for why the number of genes (up/down) is the same in the 3 groups?

5/ The co-culture model used may have some limitations, therefore a more physiological co-culture model with endothelial cells, monocytes/macrophages, and smooth muscle cells may be more relevant to delineate mechanisms of the aortic valve disease.

Overall, the manuscript is well-written with a clear methodology and the results are well discussed.

Round 2

Reviewer 2 Report

Comments and Suggestions for Authors

All my concerns were well addressed. No further comments. Thank you!